# A Cross-Sectional Study on the Prevalence of Depressive Symptoms and Its Associated Sociodemographic Factors in Peru during the COVID-19 Pandemic

**DOI:** 10.3390/ijerph192114240

**Published:** 2022-10-31

**Authors:** Angel Christopher Zegarra-López, Brian Florentino-Santisteban, Jorge Flores-Romero, Ariana Delgado-Tenorio, Adriana Cernades-Ames

**Affiliations:** 1Faculty of Psychology, Universidad de Lima, Lima 15023, Peru; 2Grupo de Investigación en Psicología, Bienestar y Sociedad, Instituto de Investigación Científica, Universidad de Lima, Lima 15023, Peru

**Keywords:** mental health, depression, Peru, COVID-19, sociodemographic factors

## Abstract

The present study aims to analyze the prevalence of depressive symptoms and its sociodemographic-associated factors in Peruvian adults. Data was extracted from a nation-wide representative survey in which depression symptoms were measured with the PHQ-9 and sociodemographic information was extracted from household data. Depression severity rates were estimated for each symptom, and responses were modeled through the Rating Scale Model to obtain a depression measure used as dependent variable on a Generalized Mixed Linear Model. The most frequent depression symptoms were emotional, such as discouragement, sad mood, hopelessness, and lack of pleasure when doing activities. Our model showed that, after controlling the effects of all the variables considered, the most relevant predictors were gender, education level, physiographic region, age, marital status, and number of coresidents. Higher depression levels were found in women, people who did not complete higher education, participants living in the Highlands, older adults, single participants, and people living alone. Thus, interventions to promote or prevent depression severity during similar situations as the pandemic should focus on specific sociodemographic groups and their particular needs.

## 1. Introduction

The sudden outbreak of the Coronavirus disease (COVID-19) imposed an unprecedented challenge for countries across the world. After being traced back to December 2019 in Wuhan, China, the disease rapidly spread and led the World Health Organization to declare it as a pandemic in March 2020 [1]. As of 30 January 2022, over 370 million cases and more than 5.6 million deaths have been confirmed globally [2]. Nevertheless, the actual impact of the pandemic is unmeasurable since its scope involves social, economic, educational and health aspects. Mental health is one of the most affected by the repercussions of the pandemic and the strict lockdown measures declared by most governments [3,4]. The uncertainty associated with the novel disease and its high mortality rate, which was constantly exposed by the media, yielded an increasing sense of fear and anxiety in the general population [5]. Additionally, social isolation and loneliness during the pandemic were linked to an increase in depression and stress-related symptoms [6]. These findings are consistent in multiple settings [7,8,9] and this implies the need to address and intervene in regards to mental health [10]. 

Despite an undeniable prevalence of mental health conditions, recent findings on their associated factors show a disproportion in the impact of COVID-19 based on sociodemographic characteristics of the individuals [11,12,13]. For instance, studies concluded that a low income or having a low socioeconomic status is a risk factor for suffering higher levels of depression and psychological stress [14,15,16,17]. Likewise, people who belong to countries with a low Human Development Index (HDI), an indicator of the World Health Organization that involves per capita income, access to education and life expectancy, presented a higher prevalence of anxiety and depression than those with a higher HDI [3]. In addition, previous research has suggested that mental health conditions are more frequent on specific demographic groups regarding gender, age and educational level [4,18,19,20,21]. Among all mental health conditions, this study focuses on the sociodemographic factors associated with depression due to its well documented impairing effects on behavior on previous research [21,22]. Depression is defined as a psychopathologic mood disorder with different severity levels that depend on the number of experienced symptoms and the degree of impairment [23]. Specific findings on the prevalence of depression during the pandemic show that, even if an individual does not have a severe pathology, most people have experienced the symptoms at some degree, such as lack of motivation, sadness, difficulty to concentrate on tasks, sleeping and eating disorders [24,25,26,27]. 

One limitation of the generalizability of these studies is the lack of research conducted in developing countries. In the case of Latin America, only a few studies have focused on the effect of the pandemic on mental health [28,29,30,31], even though the South American continent is the most affected by COVID-19 to date, with higher rates of infections and deaths [32]. Among them, Peru had the highest COVID-19 death rate in 2021 [33] and recent research shows evidence of the prevalence of depressive symptoms among the general population [34]. For instance, the Peruvian government implements the yearly Demographic and Health Family Survey, which is a study that includes the Patient Health Questionnaire-9 (PHQ-9), a measure of depression symptoms. Nevertheless, national reports on the study results do not emphasize depression. We present, in Figure 1, the tendencies of the PHQ-9 mean total score from 2017 to 2020, after weighting the data to expand the results to the population level.

The tendencies clearly show an increase in the PHQ-9 mean total score during 2020. This result is congruent with other studies conducted on Peruvian population that conclude an increase in the burden of depressive symptoms during the COVID-19 pandemic in Peru compared to previous years [34]. As the literature shows that sociodemographic factors have a strong influence on how the pandemic affected mental health, there is a need to expand previous prevalence studies in Latin American countries, such as Peru, by incorporating associated demographic variables that allow for the identification of disparities in specific groups. For this reason, the aim of the present study was to analyze sociodemographic-related factors of depression using national large-scale assessment data collected by the National Institute of Statistics and Informatics (INEI) of Peru.

## 2. Materials and Methods

### 2.1. Sample and Procedures

The National Institute of Statistics and Informatics of Peru annually implements several large-scale surveys on different subjects, such as economics, education, and health. Among them, the Demographic and Health Family Survey (ENDES) is implemented as part of a world-wide project known as Monitoring and Evaluation to Assess and Use Results Demographic and Health Surveys (MEASURE DHS) and consists of the *Household, Individual* and *Health* questionnaires, each of which has its own target population. A two-stage sampling procedure to ensure national, departmental and area representativeness was delimited, resulting in a sample of 37,390 households with a response rate of 94.1%. Data was collected between January and December of 2020 through telephone calls as the initial strategy in response to the social isolation regimes imposed at that time and then switched to face-to-face interviews on the last months. 

The results were organized in multiple datasets containing information about several demographic and health indicators. Data is available at http://iinei.inei.gob.pe/microdatos/ (accessed on 1 March 2021). The present study employs these results as secondary data. Among all the datasets, *CSALUD01* contained the main indicators of depression and sociodemographic variables. The original dataset was modified by only selecting the sociodemographic variables designated in the study, along with household variables from the *RECH0* and *RECH23* datasets. Data manipulation procedures were programmed in R with the *tidyverse* collection of packages. To ensure reproducibility, the R code is available in the supporting information as the Appendix A. All ethical considerations during data collection were ensured by the INEI. The present study was presented to the Research and Ethics Committee of the Faculty of Psychology of the Universidad de Lima and was approved in September 2021. 

### 2.2. Population and Selection Criteria

The population studied was defined as adolescents and adults who resided in Peru in 2020, during the COVID-19 pandemic. Since each questionnaire in ENDES had its own target population, a selection process was implemented based on the following inclusion criteria. First, only individuals who participated in the Health Questionnaire were considered. Second, participants who did not complete the questionnaire were excluded. Third, any participant who did not have information about the sociodemographic variables was excluded. The resulting sample consisted of 30,728 participants, from which 47.17% were men and 52.83% women, and an age range of 15 to 97 years old (M = 39.18; SD = 16.22). The resulting sample had similar sociodemographic characteristics as the population during 2020, where 49.63% were men and 50.37% were women, with most people aged between 15 to 59 years old according to the national reports during 2020. A detailed description of the sociodemographic characteristics of the sample is presented in Table 1.

### 2.3. Measures

#### 2.3.1. Depression (Patient Health Questionnaire-9)

The Patient Health Questionnaire-9 (PHQ-9) is a psychometric instrument proposed by Spitzer et al. (1999) as a self-reporting alternative for the Primary Care Evaluation of Mental Disorders (PRIME-MD). Specifically, the PHQ-9 [35] corresponds to the depression module of the PHQ and is composed of nine items that address each of the diagnostic criteria for this disorder, established by the American Psychiatric Association in the fourth edition of the *Diagnostic and Statistical Manual of Mental Disorders*. The items are presented on a Likert-type response scale that quantifies the frequency in which the participant experienced a symptom associated with depression in the last two weeks. The response alternatives are *Not at all* (0), *Several days* (1), *More than half the days* (2), and *Nearly every day* (3). The total scores of the questionnaire can be used to categorize individuals according to the level of severity of the depressive symptomatology presented [36]. The five levels proposed for this interpretation are: *Minimal* (0–4), *Mild* (5–9), *Moderate* (10–14), *Moderately severe* (15–19), and *Severe* (20–27). The psychometric properties of the PHQ-9 have been extensively studied in the literature; for instance, in heterogeneous and cross-cultural samples, a high degree of reliability has been found; however, the internal structure presents findings that propose one-dimensional and two-dimensional solutions, in both cases supported by evidence on the invariance of the measurement [37]. In Peru, a study of 30,449 participants of the ENDES 2019 identified a high degree of reliability ω = 0.87, with psychometrics being higher than the standard recommended 0.70 cutoff for reliability [38], and an acceptable adjustment to a one-dimensional structure CFI = 0.936; RMSEA = 0.089; SRMR 0.039 [39].

#### 2.3.2. Sociodemographic Variables

Sociodemographic variables that were shown to be relevant predictors of depression or mental health conditions in previous studies were included in the main model. As categorical variables, we included *gender*, a dichotomous indicator with *male* and *female* levels; *education level* defined as the latest completed academic grade considering *Primary school*, *Secondary school*, *Higher education (Non-university [NU]),* and *Higher education* (*University[U])*; *marital status* defined as a dichotomous variable differentiating whether the participant is *married*/*living* together or *single*/*not living together*; *area* as a dichotomous indicator of the *urban* or *rural* residing areas, which are classified based on the amount of people residing on a populated center (≥2000 for urban and <2000 for rural); *physiographic region,* which differentiates three main geographic units distinguished by their natural ecosystems *Coast*, *Highlands* and *Jungle*; *first language* as a dichotomous indicator with *Spanish* and *Native* as levels. We included as quantitative variables the *age* of the participants, measured in years; *coresidents* as the number of people with whom the individual cohabited during the lockdown period; and the *wealth index*, an indicator analogous to the socioeconomic level constructed by the INEI from a specialized questionnaire on the characteristics of the household and with the principal components analysis methodology. This last indicator also allows the population to be categorized into five quintiles: Low, Lower-middle, Middle, Higher middle, and High. 

### 2.4. Analysis

A four-stage analysis was carried out. First, descriptive statistics for the PHQ-9 items were analyzed to determine the prevalence rates of each depressive symptom in the main population. Second, the total score of the PHQ-9 was assessed to classify each participant according to the level of severity of the depressive symptomatology presented. Absolute and relative frequencies were estimated for the whole population and for each relevant sociodemographic variable. Third, the Rating Scale Model was used to model responses to the PHQ-9 using the Marginal Maximum Likelihood Estimation method for item calibration and expected a posteriori for personal measures. Estimations were used in mean comparisons between sociodemographic groups. Rasch models allow researchers to work with invariant interval-level measurement if the data fit the model [40]. Infit and outfit fit indicators were observed to confirm that the responses to the items fit the prescribed framework of the Rasch model, considering values of 1.40 [41] for both indicators as cut-off criteria. Finally, a random intercept mixed linear model was used to determine the strength of the predictive relationship of each of the sociodemographic variables on the measures of depression achieved. Twenty-five clusters were considered for the random intercepts model and were defined as 24 political-administrative departments and one constitutional province, because government regulations regarding the pandemic were different in each administrative cluster. All analyses considered a significance level of α = 0.05. To ensure reproducibility, the R code for data analysis procedures is available in the supporting information as the Appendix A.

## 3. Results

The prevalence of each symptom was measured through the PHQ-9 and are presented in Figure 2. Item responses were dichotomized to represent the absence or occurrence of a symptom. Emotional symptoms were the most frequent in the observed sample; specifically, 38.5% felt discouragement, sad mood, and hopelessness; and 31.5% experienced anhedonia, little interest or pleasure when doing activities. Physiological symptoms were the second most frequent since 27.3% identified difficulties in falling or staying asleep or sleeping excessively, 23.7% experienced tiredness or low energy; and 19.3% reported poor appetite or a tendency to overeat. Regarding cognitive and motor symptoms, 10.9% of cases had negative thoughts about themselves, 17.4% experienced problems concentrating on activities, and 14.9% presented catatonia at both poles. The least frequent symptom was suicidal or self-harm thoughts, only experienced by 6.8% of the sample.

Table 1 summarizes the quantity and percentage of participants in each sociodemographic group along with their distribution according to the five levels of depression severity. Everyone was categorized based on the number of symptoms reported in the PHQ-9. The vast majority show no signs of depression (78.7%) or at a mild severity (15.6%) even after disaggregating the sample into sociodemographic subgroups. With regards to gender, women show more cases of mild to severe depression compared to men. In educational level, a tendency to show a higher percentage of people without signs of depression increases as a higher level of education is achieved. 

Single participants had higher rates of mild to severe depression than married or participants who live with their partners. With respect to wealth, people on the lower quintiles experience more severe depressive symptoms than the ones on the higher quintiles. Additionally, as a participant report having more coresidents it presents a tendency of lower mild to severe depression rates. People who lived in the Highlands reported higher levels of mild to severe depression than the ones who live in the Coast or Jungle. Participants whose first language was Spanish had lower rates of mild to severe depression than participants with native first language. No pattern was found regarding area.

To estimate mean differences, responses were modeled through the unidimensional Rating Scale Model. A Confirmatory Factor Analysis using polychoric correlations and the Variance Adjusted Weighted Least Squares (WLSMV) estimator provided fair evidence of a single dimension being measured χ^2^(27) = 4563.187, *p* < 0.001; CFI = 0.977; TLI = 0.969; RMSEA = 0.074 (90% CI: 0.072–0.076); SRSMR= 0.050. Structure thresholds were ordered as expected and all items presented an adequate fit to the Rasch model (Infit = 0.890–1.341; Outfit = 0.712–0.923). The empirical reliability was acceptable for the empirical research Rp = 0.727. 

Figure 3 presents bivariate analyses as differences in depression measures by each sociodemographic group selected for this study. Statistically significant differences were found by comparing men (M = −0.152) and women (M = 0.136), t(30,719) = −25.67, *p* < 0.001, where women presented higher levels of depression with a small effect size d = −0.292. In the same way, differences were found between at least a few groups in regards to the reported educational level F(3,30724) = 64.76, *p* < 0.001, ω^2^ = 0.006. Multiple comparisons with Tukey tests indicated that there is a higher level of depression in people with complete primary versus complete secondary *p* < 0.001, d = −0.146; technical superior *p* < 0.001, d = −0.216; and higher university studies *p* < 0.001, d = −0.204. A higher level of depression was also found in individuals with completed high school compared to technical higher education *p* < 0.001, d = −0.073, and university higher education *p* = 0.002, d = −0.062, but with a negligible effect size in both cases. No statistically significant differences were found between people with higher technical and university studies *p* = 0.952, d = 0.012. 

Regarding the physiographic region strata, statistically significant differences were found between groups F(2,30725) = 92.22, *p* < 0.001, and ω^2^ = 0.006. Multiple comparisons indicated a higher level of depression in individuals from the Highlands when comparing them with the Coast *p* < 0.001, d = 0.165 and Jungle *p* < 0.001, d = −0.149, but in both cases, there were irrelevant effect sizes. No differences were found between Coast and Jungle *p* = 0.461, d = −0.017. 

Analyses between wealth index quintiles showed statistically significant differences F(4,30723) = 12.29, *p* < 0.001, ω^2^ = 0.001. Specifically, differences denote a lower level of depression reported by individuals from the fifth (highest) wealth quintile when compared to the first *p* < 0.001, d = −0.119, second *p* < 0.001, d = −0.143, third *p* < 0.001, d = −0.117 and fourth *p* < 0.001, d = −0.134; however, the effect size is negligible in such comparisons. 

Comparisons with respect to the area indicate that there are no statistically significant differences between urban (M = 0.004) and rural (M = −0.007) strata, t(21,315) = 0.92, *p* = 0.358, d = 0.011. The number of co-residents implied statistically significant differences F(2,30725) = 61.74, *p* < 0.001, ω^2^ = 0.004. Post-hoc studies reveal differences between all levels; for instance, people who live alone compared to people with three or more coresidents *p* < 0.001, d = −0.121; or comparing three or more cohabitants with four or more cohabitants *p* < 0.001, d = −0.101. However, when comparing individuals who live alone versus individuals who live with four or more people, a higher level of depression was found for the former with a small effect size *p* < 0.001, d = −0.226. 

Statistically significant differences were identified in marital status when comparing participants who were married or living with their partners (M = −0.130) and single, widowed or divorced (M = −0.083), with a higher level of depression reported by the second group t(21267) = −8.79, *p* < 0.001, d = −0.106. Age groups showed statistically significant differences F(3,30724) = 180.30, *p* < 001, ω^2^ = 0.017. 

In general, the highest level of depression was reported by older adults aged 60 years or more compared to young people *p* < 0.001, d = 0.331, adults from 25 to 39 years old *p* < 0.001, d = 0.382 and adults from 40 to 59 years *p* < 0.001, d = 0.171. The second group with higher levels of depression were adults aged 40 to 59 years compared to young people *p* < 0.001, d = −0.154 and adults aged 25 to 39 years *p* < 0.001, d = 0.201. 

Finally, it was found that young people presented higher levels of depression than adults aged 25 to 39 years, but with an effect size without practical relevance *p* < 0.001, d = −0.047. Statistically significant differences were found when comparing individuals whose first language was Spanish (M = −0.049) in contrast to those whose first language was a native language (M = 0.161), t(113,270) = −15.14, *p* < 0.001, d = −0.210.

Further exploration on how sociodemographic covariates are related to the measure achieved in depression are examined in a random intercepts linear mixed model to allow controlling the effects of other variables. The political-administrative department variable with 25 constituencies throughout Peru was considered as a clustering variable since the political mandates against the pandemic were different in each of these clusters. The results of the model are presented in Table 2.

The variance of random effects was 0.037, denoting low variability levels between the 25 clusters. The explained variance by the fixed effects and the total model were s R^2^_marginal_ = 0.048 and R^2^_conditional_ = 0.085, respectively. No multicollinearity was found. After controlling the effect of the department, most variables proved to be statistically significant predictors of depression. As expected, a statistically significant difference regarding gender was found, showing that women had higher levels of depression. The primary education level had lower levels of depression compared to higher education. Being single implied higher depression than being married. The number of coresidents had a negative statistically significant relationship with depression, while age had a positive relationship; no statistically significant differences were found with respect to first language or wealth index.

## 4. Discussion

The COVID-19 pandemic had a devastating impact on mental health, as evidenced in recent scientific literature. Several studies report the prevalence of mental health conditions, mainly depression, anxiety, and stress [3,4]. Nevertheless, the impact is disproportional regarding sociodemographic strata such as gender, socioeconomic status, or educational level [11,12,13]. Studies on sociodemographic associated factors allow researchers and policy makers to gain a deeper understanding on the impact of the pandemic as well as to identify relevant disparities on specific groups of the population. For this reason, the aim of this study was to explore sociodemographic associated factors of depression on Peru, one of the most affected countries by the pandemic. 

Regarding specific depression-related symptoms, emotional and physiological symptoms like sad mood, hopelessness, lack of energy or sleep disorders were the most frequent. As previous research shows, lockdown measures brought sudden changes in people’s lifestyles, including social isolation, death of friends and relatives, and unemployment which have been linked to an increased prevalence of the above-mentioned symptoms [42,43,44]. Motor and cognitive impairing symptoms like negative thoughts towards oneself, problems concentrating, or catatonia were less frequent, though its presence in more than 10% of the sample should be noted due to the incapacitating nature of these symptoms [21]. Lastly, suicidal or self-harm thoughts were the least frequent symptoms, although there was an alarming rate of 6.8%. This is an alarming percentage, considering that studies before COVID-19 on Peru showed a peak of a 1.95% suicide rate on 2019 [45] and suicidal behavior is one of the main predictors of suicide [46]. Modern findings show that people with a higher risk of suicide tend to experience social isolation, feelings of loneliness and other psychiatric disorders, such as anxiety, stress, and depression, all which have been increasing due to COVID-19 [47,48,49]. Moreover, there is plenty of evidence of an increase in suicide-related behaviors, denoting the need for developing intervention and prevention strategies [50]. 

With respect to depression severity and main sociodemographic differences, we found that approximately two out of ten participants had a mild to severe depression on the total sample. After disaggregating and controlling results by sociodemographic strata, we found statistically significant differences regarding gender; specifically, women experience more severe depression symptoms than men. This finding is consistent with previous studies regarding gender differences in mental health during the pandemic that targets women as a more vulnerable group [51,52,53]. Explanations could be linked to social studies in Latin American countries that found that women had to withdraw their occupations to take on household chores, generating disparities compared to men [54]. Likewise, the findings suggest an increase in domestic violence against women during the pandemic [55,56,57], lockdown measures contribute to spending more time at home with their abuser and the restriction of physical contact with other people limits social support [55,58] and increases the prevalence of depressive symptoms [52]. 

Having a higher educational level is related to lower levels of depression. Participants who completed primary or secondary school presented higher levels of depression compared to those with higher technical and university education. Similar findings were reported according to educational level in different settings [59,60,61,62]. Explanations of this relationship rely on how education grants access to a better quality of life by allowing access to better job opportunities and higher socioeconomic status [63]. 

Our findings suggest that having a partner or being married is a significant protective factor against depressive symptomatology, as has been commonly reported in the literature [64]. It has been shown that being married has a positive relationship with mental health due to the emotional and economic support needed during the COVID-19 pandemic [65]. Nevertheless, when a couple shares a poor relationship, it might even increase the risk of suffering from depressive symptoms [66]. Thus, just being married does not guarantee a positive impact on mental health, since the quality of the relationship is also important. 

On a similar topic, we found that living with more people tends to reduce depression levels; additionally, more than 25% of people who lived alone reported a mild to severe depression. Cultural and personal differences on collectivism or individualism might be related to psychological responses to the pandemic [67]. How spending time with friends and family is valued in some cultures could result in an increase or decrease in psychological well-being and depressive symptoms [68,69]. As evidence, several studies found mixed results regarding the relationship between member cohabiting during the pandemic and mental health conditions [70,71,72], since other cultural factors or even quality of the relationships might be intervening [67,73]. 

The results indicate no significant differences in depressive symptoms between rural and urban areas, which contradicts previous findings that suggested a greater depressive severity in urban areas [74,75]. One explanation for this phenomenon is that the settings on which studies were carried out differ a lot from Peru. In Peru, rural areas tend to have limitations towards accessing essential services and resources and report higher poverty levels. For this reason, people from rural areas had more difficulty coping with the pandemic and its impact. Likewise, when analyzing the results by physiographic region, we found that people from the Highlands present higher severity levels of depression than people from Coast or Jungle. As mentioned before, inequalities in Peru on essential services and access to resources might explain this result since the Highlands tend to have many limitations in these aspects, whereas the Coast characterizes for plenty of access to essential needs, which is facilitated because of the means of transport and communication established in the territory. However, the Jungle also lacks access to social services, but differences might be explained because lockdown measures were not strictly imposed on several cities of the Jungle; thus, no relevant changes were observed for them. 

With respect to age, the results indicate that the older a person is, the higher the levels of depressive indicators are. These results are consistent with previous studies conducted on similar age groups [76,77]. Nonetheless, these findings contradict several studies where it is stipulated that young adults report higher depression levels compared to older age groups due to the way in which the pandemic affected their education and social-oriented lifestyles [78,79,80,81]. Nonetheless, systematic reviews suggest that the nature of the impact of the pandemic on mental health on older adults is more complex, since the capacity of older adults to adapt to adversities depend on cultural, social, economic, and individual factors, the impact of COVID-19 on mental health in this population group is expected to vary across countries and is yet to be evaluated [72]. Another possible explanation is the high mortality rate in Peru, especially in older adults and the precarious conditions of health services for the elderly. 

Additionally, lockdown measures and social distancing have affected a large number of elderly people who depend on social support centers for their care, such as community care centers, social assistance, a volunteer group, or asylum in places of worship, limiting the support networks of the elderly, canceling important activities, such as spending time with family and close friends, which has had considerable consequences on their mental health since loneliness and isolation have been shown to have a positive connection with depression for this group [78,82,83]. 

Our study revealed that people who did not have Spanish (23.2%) as their mother tongue have higher levels of depression compared to people whose first language was Spanish. Nevertheless, after controlling for other factors, differences were not found. Most people whose first language is not Spanish tend to be from rural areas with limited access to resources and essential services that lead them to migrate to urban areas at a latter stage of their lives. Studies in Europe show that immigrants have greater depressive symptoms than native Europeans and economic, social and cultural exclusion [84]. In Peru, centralism in urban cities closes opportunities to improve mental health and promote migration from provinces to more developed regions and with more resources for the treatment of mental illnesses such as depression. Regardless, the main reason for the precarious attention in mental health for non-Spanish speakers is the language barrier between patients approaching care and doctors, nurses and medical staff, proof of this is found even in developed countries [85]. Therefore, in Peru, where the majority of the population speaks Spanish, it is difficult for native language speakers to fully integrate into the health system without the help of competent translators who can transcend language barriers in medical care, which would increase the degree of uncertainty and feelings of exclusion in this population. 

Regarding wealth, after controlling the effect of other variables, no relationship was found, though participants from the highest wealth quantile differed significantly from others, showing lesser depression levels. No substantial differences were found between the other quintiles. This reaffirms the results found in contemporary research, which denotes that a higher wealth or economic status is considered as a protective factor against depression and other mental health problems [15,16,86,87]. A low income is associated with a greater number of negative life events, access to essential services and unemployment, which increases the possibility of suffering some disorders, including depression [88,89]. 

These results suggest that interventions on mental health promotion and prevention during similar conditions as the ones experienced during the COVID-19 pandemic should focus on specific sociodemographic groups and their own needs. For instance, women need specific interventions to prevent potential domestic violence, increase social support and to promote their continuity at their jobs instead of abandoning them to take care of the housework. Regarding age, interventions should focus on older adults since lockdown measures limit their social networks, especially for the ones depending on community care centers or asylums. Remote strategies for strengthening their networks could be considered to reduce feelings of loneliness and isolation [90]. Furthermore, the Peruvian government should recognize that the Highlands region reported the higher levels of depression and consider assigning more resources to address the lack of access to essential services during the lockdown periods. Lastly, people who end up living in isolation and far away from their relatives are at a higher risk of severe depression, interventions should consider strengthening their social networks, and in lockdown situations, means of remote communication can be a solution [91].

### Limitations

It is important to recognize that this study has some limitations. One of these is the representativeness of the sample. Even though the ENDES study had a complex sampling design to ensure generalizability to all the population, we did not consider such design during the analysis phase since we had to gather data from different questionnaires, each with its own complex design. Nevertheless, the resulting sample after the data inclusion criteria had similar sociodemographic characteristics as the whole population during 2020, regarding gender and age. Another limitation is the lack of COVID-19 related variables. It has been shown that the pandemic had a strong impact on mental health [92] and clarifying how each participant was affected by the pandemic could have enriched the discussion and the results after isolating the effect of COVID-19. Since we used a secondary data, we could not control which questions were made available at the survey and could only work with the ones that were already included. We encourage further research to focus con COVID-19 specific effects on mental health. To end with, we chose to focus on sociodemographic factors of the year 2020 since the COVID-19 outbreak and lockdown measures were at its peak; nevertheless, ENDES is an annually implemented survey and longitudinal approaches may bring important insights on the fluctuation of depression levels during the last years, we encourage future studies to implement such longitudinal approaches. 

## 5. Conclusions

After controlling the effects of all variables considered, the most significant predictors were gender, with higher depression demonstrated by women. Regarding education level, lesser depression symptoms are shown for participants who completed higher education. Physiographic region analyses indicate that higher depression levels are found for people who reside in the Highlands. A positive relationship was found between age and depression, indicating an increase in depression as a person gets older. Being married acts as a protective factor against depression. The number of coresidents indicate that living with more people tends to be related to a lower level of depression severity. No substantial differences were found regarding wealth or first language. 

## Figures and Tables

**Figure 1 ijerph-19-14240-f001:**
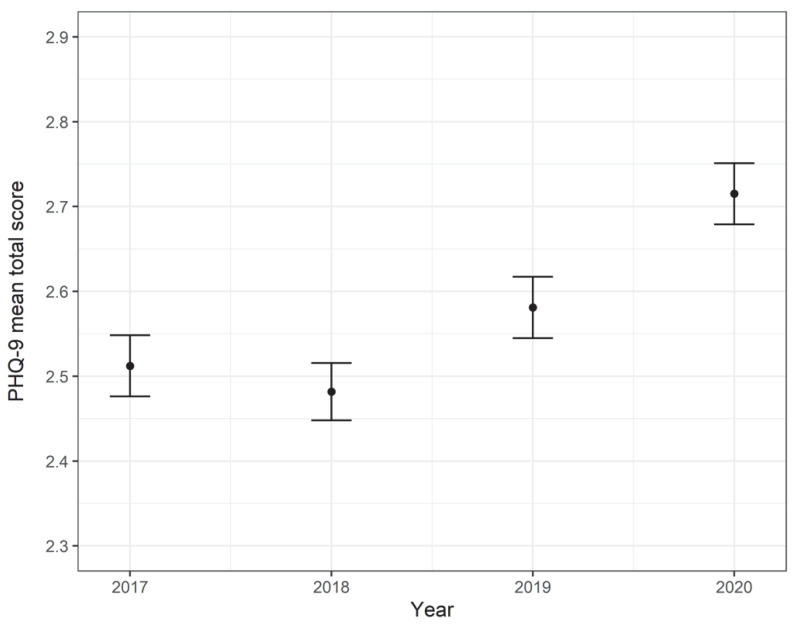
PHQ-9 mean total score tendencies from 2017 to 2020.

**Figure 2 ijerph-19-14240-f002:**
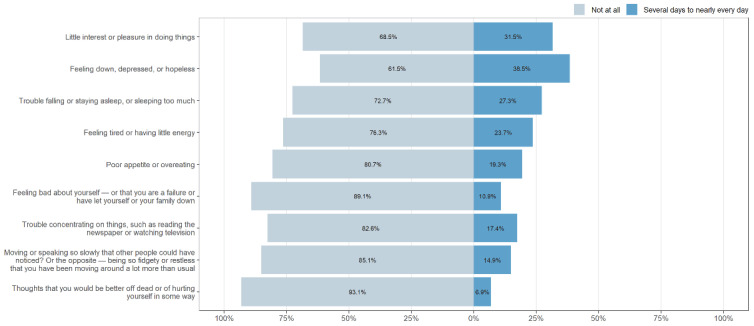
Prevalence of depression symptoms.

**Figure 3 ijerph-19-14240-f003:**
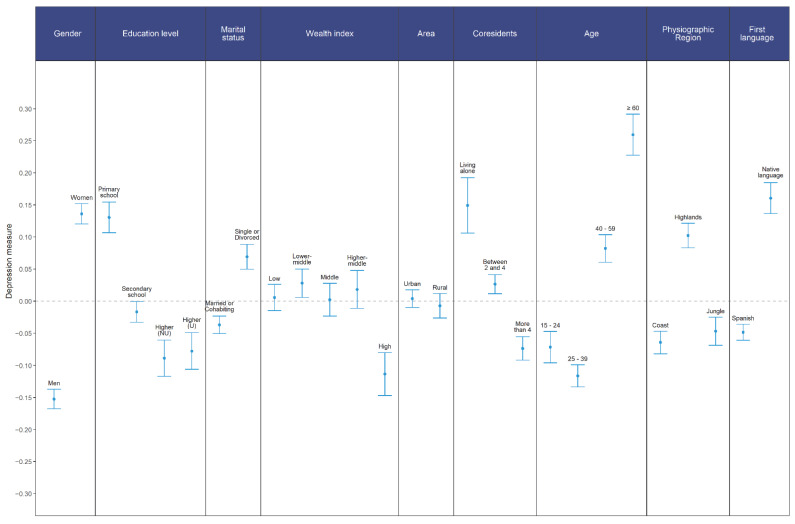
Depression mean differences between sociodemographic groups.

**Table 1 ijerph-19-14240-t001:** Depression severity by sociodemographic groups.

Sociodemographic Group		Group Size n (%)	Depression Severity				
			None (PHQ ≤ 4)	Mild (PHQ ≤ 9)	Moderate (PHQ ≤ 14)	Moderately Severe (PHQ ≤ 19)	Severe (PHQ > 19)
Total		30,728 (100.0%)	24,169 (78.7%)	4803 (15.6%)	1070 (3.5%)	451 (1.5%)	235 (0.8%)
Gender	Men	14,494 (47.2%)	12,170 (84.0%)	1767 (12.2%)	336 (2.3%)	152 (1.0%)	69 (0.5%)
	Women	16,234 (52.8%)	11,999 (73.9%)	3036 (18.7%)	734 (4.5%)	299 (1.8%)	166 (1.0%)
Educational level	Primary school	7475 (24.3%)	5492 (73.5%)	1400 (18.7%)	338 (4.5%)	153 (2.0%)	92 (1.2%)
	Secondary school	14,452 (47.0%)	11,493 (79.5%)	2185 (15.1%)	465 (3.2%)	215 (1.5%)	94 (0.7%)
	Higher education (NU ^1^)	4435 (14.4%)	3614 (81.5%)	621 (14.0%)	139 (3.1%)	39 (0.9%)	22 (0.5%)
	Higher education (U ^2^)	4366 (14.2%)	3570 (81.8%)	597 (13.7%)	128 (2.9%)	44 (1.0%)	27 (0.6%)
Marital status	Married	20,013 (65.1%)	15,969 (79.8%)	3004 (15.0%)	649 (3.2%)	251 (1.3%)	140 (0.7%)
	Single	10,715 (34.9%)	8200 (76.5%)	1799 (16.8%)	421 (3.9%)	200 (1.9%)	95 (0.9%)
Wealth index quintile	Low	9482 (30.9%)	7387 (77.9%)	1546 (16.3%)	323 (3.4%)	139 (1.5%)	87 (0.9%)
	Lower-middle	7812 (25.4%)	6113 (78.3%)	1270 (16.3%)	261 (3.3%)	109 (1.4%)	59 (0.8%)
	Middle	5899 (19.2%)	4651 (78.8%)	898 (15.2%)	209 (3.5%)	100 (1.7%)	41 (0.7%)
	Upper-middle	4339 (14.1%)	3392 (78.2%)	680 (15.7%)	179 (4.1%)	58 (1.3%)	30 (0.7%)
	High	3196 (10.4%)	2626 (82.2%)	409 (12.8%)	98 (3.1%)	45 (1.4%)	18 (0.6%)
Area	Urban	20,168 (65.6%)	15,891 (78.8%)	3104 (15.4%)	713 (3.5%)	309 (1.5%)	151 (0.7%)
	Rural	10,560 (34.4%)	8278 (78.4%)	1699 (16.1%)	357 (3.4%)	142 (1.3%)	84 (0.8%)
Coresidents	Living alone	2309 (7.5%)	1691 (73.2%)	419 (18.1%)	109 (4.7%)	62 (2.7%)	28 (1.2%)
	Between 2 and 4	17,480 (56.9%)	13,609 (77.9%)	2826 (16.2%)	630 (3.6%)	275 (1.6%)	140 (0.8%)
	More than 4	10,939 (35.6%)	8869 (81.1%)	1558 (14.2%)	331 (3.0%)	114 (1.0%)	67 (0.6%)
Age	15–24	5958 (19.4%)	4846 (81.3%)	849 (14.2%)	170 (2.9%)	67 (1.1%)	26 (0.4%)
	25–39	11,839 (38.5%)	9748 (82.3%)	1616 (13.6%)	296 (2.5%)	112 (0.9%)	67 (0.6%)
	40–59	8753 (28.5%)	6630 (75.7%)	1516 (17.3%)	374 (4.3%)	151 (1.7%)	82 (0.9%)
	≥60	4178 (13.6%)	2945 (70.5%)	822 (19.7%)	230 (5.5%)	121 (2.9%)	60 (1.4%)
Physiographic region	Coast	12,301 (40.0%)	9924 (80.7%)	1709 (13.9%)	408 (3.3%)	176 (1.4%)	84 (0.7%)
	Highlands	11,101 (36.1%)	8337 (75.1%)	1985 (17.9%)	447 (4.0%)	213 (1.9%)	119 (1.1%)
	Jungle	7326 (23.8%)	5908 (80.6%)	1109 (15.1%)	215 (2.9%)	62 (0.8%)	32 (0.4%)
First language	Spanish	23,591 (76.8%)	18,901 (80.1%)	3457 (14.7%)	775 (3.3%)	307 (1.3%)	151 (0.6%)
	Native language	7137 (23.2%)	5268 (73.8%)	1346 (18.9%)	295 (4.1%)	144 (2.0%)	84 (1.2%)

^1^ Non-University. ^2^ University.

**Table 2 ijerph-19-14240-t002:** Random intercept linear mixed model.

Variable	$\hat{\beta }$	95% CI		SE	df	t	p
Gender ^1^	0.306	0.284	0.328	0.011	30,697.618	27.598	<0.001
Education level ^2^ (Secondary)	−0.012	−0.043	0.020	0.016	30,715.312	−0.729	0.466
Education level (Higher NU ^3^)	−0.091	−0.132	−0.050	0.021	30,709.807	−4.357	<0.001
Education level (Higher U ^4^)	−0.115	−0.158	−0.071	0.022	30,715.079	−5.151	<0.001
Physiographic region ^5^ (Highlands)	0.045	−0.008	0.098	0.027	3719.722	1.649	0.099
Physiographic region (Jungle)	0.101	0.033	0.170	0.035	2298.664	2.888	0.004
Wealth index	0.005	−0.011	0.021	0.008	30,557.854	0.613	0.540
Coresidents	−0.013	−0.019	−0.007	0.003	30,715.018	−4.222	<0.001
Marital Status ^6^	0.136	0.113	0.160	0.012	30,697.847	11.433	<0.001
Age	0.008	0.008	0.009	0.000	30,703.178	21.071	<0.001
First language ^7^	0.032	−0.001	0.066	0.017	27,864.642	1.888	0.059

^1^ Men as reference group. ^2^ Primary education level as reference group. ^3^ Non-University. ^4^ University. ^5^ Coast as reference group. ^6^ Married as reference group. ^7^ Spanish as reference group.

## Data Availability

Data is available http://iinei.inei.gob.pe/microdatos/ (accessed on 1 March 2021).

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
