# Peer review of "A Cross-Sectional Study on the Prevalence of Depressive Symptoms and Its Associated Sociodemographic Factors in Peru during the COVID-19 Pandemic"

_ijerph, 2022, doi:10.3390/ijerph192114240_

Round 1

Reviewer 1 Report

Thank you for your submission your important works. I can understand that your works is the first reports about depressive state among Perusian under the COVID-19 outbreak. Through my reviewing your manuscript, I would make several comments to be rivised.

1. Novelty in this study: I can understand that this study is first report regarding the depressive status in Latin Ameria during the COVID-19 outbreak, however, this study have conducted annually, it is not purpose of investigation for psychological distress affected by the COVID-19 outbreak. I think it is critical issues as a regular article to be published.

2. Representative of the sample: Selected subjects were almost equivalent to national population? Could you show the difference(e.g., ages, gender) compared with the this subjects and national average, and discuss about the representative of samples?

3. Ethical considerations: Could you mention about the ethical considerations? If the authors are recieved an approval from the ethical committe, it needs to describe the approval No. and its date. 

4. Response rate: In order to evaluate whether this study findings are reliable and valid, it is necessary to mentioin the response rate. 

5. Limitation and strength: I think it had better to discuss about the limitation and strength in this study in detail. If the authors did not ask to subjects how affected by the COVID-19 outbreak, I think it is critical issues because it has a great gap between the title and the methodology in this study. Besides, unfortunately, I was not able to understand a clear strength in this study except for the first report in Latin America.

6. Query about the COVID-19: I was not able to query about the direct query about the COVID-19. Did the authors ask for participants about anxiety for infection or difficulty of living life due to the COVID-19 outbreak? 

7. Comparison with the pre- COVID-19 outbreak: This national survey is conducted annualy, but have the authors compared the results in this study with those before the COVID-19 outbreak? It seems that the comparison with before the outbreak will be an important consideration, and interesting matters for readers.

8. Practicality in the COVID-19 outbreak: Please consider in the discussion or conclusion section (+abstract) how the obtained findings can be applied to the actual society to prevent mental health issues.

9. Format the text: It is very tough to read and catch the contents in this manuscript because it is not divided into several paragraphs.

Author Response

Dear reviewer, 

Thank you for your observations. 

We took your suggestions and improved the manuscript.

We are attatching a table with all changes made.

Reviewer 2 Report

29.9.2022

Manuscript ID IJHS-22-0050, entitled “ A cross-sectional study on the prevalence of depressive symptoms and its associated sociodemographic factors in Peru during the COVID-19 pandemic ”

Comments to the Authors

Thank you for your description and analysis of your study aimed to examine the prevalence of depressive symptoms and its sociodemographic associated factors in Peru. 

This is a well-written manuscript reporting the results of a nationwide study in Peru.

COVID-19 pandemic constitutes an extraordinary health, social and economic global challenge. The impact on people's mental health is expected to be high, including depression and anxiety.

Addressing mental health during and after this global health crisis should be placed into the international and national public health agenda to improve citizens’ well-being. In particular, countries such as Peru faced high mortality rates due to COVID-19.

The findings could lead to change in influencing policy makers' decisions and conducting further research.

See please my comments are marked in the body of the text

Author Response

(The authors gave the same response as above.)

Reviewer 3 Report

1. The paper addresses a cross-sectional study with a high sample 30.728 of Peruvian adults. The main goal is to analyze the prevalence of depressive symptoms with a diverse number of sociodemographic factors.

2. Despite the quality of the sample and the different statistical analyses, the results and discussion are confusing.

3. It is important to review how the results are presented and discussed in order to make the article clearer and more objective.

4. For example, in results section: it is not clear if women show more cases of depression compared to men. We are talking about women, in general, or women  with other characteristics: with children?, married?, low level of education....

5. It is relevant to provide this information.

6. The results will be more objective and the discussion will be more structured

7. This change may improve the guiding thread of the paper.

Author Response

(The authors gave the same response as above.)

Round 2

Reviewer 1 Report

Thank you for your polite works to revise. Following the last comments that I pointed out, almost points were mentioned in the limitation section. However, the most significant point "the novelty in this study" has not still overcome, I assumed. 

I understand that this study was the examination of residents who showed depressive symptoms and its related factors without asking questions about COVID-19 directly, therefore, I think it is necessary to compare with status in the pre-COVID-19 outbreak at least because this survey was conducted annually.

I think that this point is a critical issue to be published in this journal regarding lack of novelty.

Author Response

Dear Reviewer, 

Thank you for your observations.

Even though the Demographic and Health Family Survey (ENDES) is yearly implemented, the Peruvian government does not emphasize depression on the national reports of results. Therefore, we estimated the PHQ-9 total score fore years 2017 to 2020 and then calculated the weighted means and its standard errors to compare depression levels pre-COVID-19. We included this as a first figure on the introduction section.

Reviewer 3 Report

Nothing to declare

Author Response

Dear reviewer,

Thank you for your time and your approval.